# Availability, price and nutritional assessment of plant-based meat alternatives in hypermarkets and supermarkets in Petaling, the most populated district in Malaysia

**Katty Jia Qi Lou**[1], **Nadia Nantheni Rajaram**[2], **Yee-How Say**[1]*

1 Department of Biological Sciences, School of Medical and Life Sciences, Sunway University, Subang Jaya, Selangor, Malaysia, 2 Sunway Centre for Planetary Health, Sunway University, Subang Jaya, Selangor, Malaysia

* yeehows@sunway.edu.my

**Data Availability Statement:** All relevant data are within the manuscript and its Supporting Information files.

## Abstract

This study investigated the availability, price, nutritional composition, and quality of plant-based meat alternatives (PBMAs) in supermarkets and hypermarkets in Petaling, the most populated district in Malaysia. The survey covered 85 stores through on-site visits, identifying 251 unique PBMA products from 42 brands. The PBMAs were categorized into nine groups: Burgers/Patties, Coated Meat, Luncheon Meat, Minced Meat, Pastries, Pieces/Chunks/Fillets/Strips, Prepacked Cooked Meals, Sausages, and Seafood Balls/Cakes/Meatballs. The survey indicated widespread availability of PBMAs in Petaling, with premium supermarkets offering the most extensive selection. The median prices of PBMAs across categories were below MYR 7 (~USD 1.6) per 100 grams, with Pieces/Chunks/Fillets/Strips being the most affordable. Nutritionally, PBMAs exhibited lower energy, total fat, saturated fat, sodium, and protein levels than meat equivalents, while showing higher carbohydrate content. Nutri-Score profiling revealed generally favourable scores (Grade A to C), indicating moderate to good nutritional quality. These findings suggest that PBMAs offer a viable alternative for consumers in Malaysia seeking healthier and more sustainable food options, particularly for those aiming to reduce their intake of fat and sodium. However, PBMAs may not be recommended for individuals seeking a higher protein and lower carbohydrate diet. Further research is recommended to explore micronutrient profiles to enhance dietary decision-making.

## Introduction

Plant-based meat alternatives (PBMAs) are designed to mimic the taste, texture, and nutritional profile of conventional meat products while being derived entirely from plant sources [1, 2]. These products, including burgers, sausages, and minced meat, have become popular among vegetarians and vegans and flexitarians—individuals who seek to reduce their meat consumption for various reasons [3]. PBMAs are often formulated with specific ingredients

**Funding:** The author(s) received no specific funding for this work.

**Competing interests:** The authors have declared that no competing interests exist.

aimed at replicating the texture and taste of meats. These ingredients typically include a blend of plant-based proteins such as legumes (e.g., soy, beans, peas) and cereals like wheat and quinoa [4]. Fats, commonly sourced from coconut oil, sunflower oil, or predominantly soy oil, are added to mimic the mouthfeel and juiciness associated with meat [4, 5]. Additionally, ingredients such as mushrooms are used for their umami flavours, enhancing taste and aroma in PBMAs [6]. To ensure product cohesion during cooking, hydrocolloids like methylcellulose and gums are also used, aiding in replicating the texture of traditional meat products [5].

The global market for PBMAs has rapidly expanded, generating USD 11.38 billion in 2024, with an expected compound annual growth rate of 10.2% from 2024 to 2028 [7]. In Malaysia, demand is also rising, with projections of 1.5 million kg in volume and USD 23.22 million in revenue by 2028 [8]. This surge is mainly driven by the increasing awareness of the health benefits associated with plant-based diets, which are linked to lower risks of diabetes, cancer, cardiovascular diseases, and overall mortality, compared to omnivorous diets often featuring red and processed meats [9, 10]. This is because plant-based foods are typically lower in saturated fats and cholesterol but higher in fibre, though they may lack certain nutrients like vitamin B12, protein, and calcium compared to meats and dairy products [9]. Beyond health concerns, environmental concerns, animal welfare, and consumer curiosity about new tastes and sensory experiences, along with the longer shelf life of PBMAs, also drive their popularity [11–13].

The growing popularity of PBMAs has sparked debates about their healthiness compared to traditional meat-based products. Most PBMA categories, including mince, meatballs, and burgers, generally have better nutritional profiles than their meat equivalents [11]. PBMAs often provide more dietary fibre and contain lower levels of calories, total fat, saturated fat, cholesterol, sodium, and salt compared to meat products [3, 9, 11]. Certain PBMAs are also fortified with vitamins and minerals commonly found in red meat, like iron and zinc, enhancing their nutritional value [10]. However, concerns exist regarding the bioavailability of nutrients in PBMAs, as the iron in meat is heme iron, which is more easily absorbed than the non-heme iron found in PBMAs [10]. Furthermore, because anti-nutrients such as phytates, lectins, and tannins may limit the absorption of vitamins and minerals, their presence in PBMAs must also be considered [3]. Critics argue that PBMAs, despite being plant-based, are classified as ultra-processed foods, containing ingredients with little to no whole food content and frequently enhanced with additional tastes, colours, emulsifiers, and other additives [1–3, 10, 11]. Consumption of ultra-processed foods is strongly linked to an increased risk of obesity and non-communicable diseases [14, 15]. While there are clear benefits and drawbacks to consuming PBMAs, their health implications vary based on the specific product, consumption frequency, and overall dietary quality [3].

In many markets, PBMAs tend to be more expensive than conventional meat products, which often hinders consumer adoption [3, 9, 16, 17]. This is because the cost of processing PBMAs can drive up their final price for consumers [9]. Research indicates that if PBMAs are less expensive, people are more likely to buy them [12]. However, another study contradicted these findings by reporting no significant price difference between PBMAs and their meat equivalents [9]. Despite variable prices, the availability of PBMAs is steadily increasing in supermarkets globally [1, 18]. In Malaysia, the plant-based food market is also expanding, offering a wide variety of products [19]. The growing availability of PBMAs in various retail environments, from upscale grocery stores to more accessible supermarkets, reflects a broader trend toward integrating plant-based options into mainstream food choices.

Despite numerous studies assessing the nutritional profile and healthiness of PBMAs [1, 2, 9, 11, 13], a quantitative analysis is currently unavailable in Malaysia. With the rise of PBMAs in Malaysia, Malaysian consumers would be interested to know whether these products are available in their neighbourhood, and how affordable are they, and how nutritious are they by

themselves or when compared with meat equivalents. Therefore, the objectives of the study were: 1. To assess the availability and price of PBMAs available in hypermarkets and supermarkets in Petaling, the most populated and one of the most affluent district in Malaysia; 2. To compare the key nutrient contents (energy, protein, total fat, saturated fat, carbohydrates, and sodium) of these PBMAs with their meat equivalents by their product categories; 3. To evaluate the nutritional profiles of these PBMAs by their product categories using the Nutri-Score Nutritional Rating System to determine their relative healthiness.

## Methods

### Study site, sample selection and collection

Petaling District is the most populated district in the country, registering the highest population in 2023 at 2.3 million [20]. This district is situated in the middle of the Klang Valley, adjacent to the capital of Malaysia–Kuala Lumpur, and has experienced tremendous urbanization. Supermarkets and hypermarkets in the Petaling District (covering Subang, Damansara, Shah Alam, Puchong, Kelana Jaya, and Petaling Jaya) were identified through Google searches. Searches such as "Supermarkets and Hypermarkets in Subang", "Supermarkets and Hypermarkets in Damansara" and henceforth, were performed, followed by on-site visits to all identified locations from April to June 2024. We performed a preliminary eyeballing survey at all small and medium-sized grocery stores, supermarkets and hypermarkets, and found that small and medium-sized grocery stores, whether independent or chain-based, offered limited amounts (less than 3) of PBMAs. Therefore, this study focused exclusively on hypermarkets, supermarkets, and their chains, excluding small and medium-sized grocery stores. Out of 85 visited locations, 81 offered PBMAs. PBMAs selected for this study were defined as: products made from plant-based ingredients that are designed to replicate meat in taste, texture, and overall experience; available in chilled, frozen, or canned forms; labelled as "vegan" or "vegetarian." Products like tofu and tempeh, which do not mimic meat, were excluded. During visits, photographs of the front packaging and nutrition labels (per 100g) of each PBMA were taken, and data on prices, packaging sizes, countries of origin, and availability were recorded.

### Product categorization

The PBMAs identified were categorized into nine groups: Burger/Patties, Coated Meat, Luncheon Meat, Minced Meat, Pastries, Pieces/Chunks/Fillets/Strips, Prepacked Cooked Meals, Sausages, and Seafood Balls/Cakes/Meatballs (S1 Table). These categories were chosen based on previous studies that have established these groups as representative of the variety and forms in which PBMAs are typically available in the market [11, 21–24]. In this study, two additional categories, Pastries and Prepacked Cooked Meals, have been included. The inclusion of these categories was because PBMA products are increasingly diverse due to growing consumer demand for more varied and convenient options. As these items are now available in Malaysia, adding these categories ensures a comprehensive analysis of all available PBMA options in the market.

 **Availability assessment.** The supermarkets and hypermarkets ($n$ = 81) offering PBMA options were categorized as hypermarkets, supermarkets, and premium supermarkets (S2 Table). Based on definitions provided by the Malaysian-German Chamber of Commerce and Industry (MGCC) [25], these categories are described as follows: Hypermarket: Large self-service stores with a sales floor area of at least 5,000 square meters, offering an extensive range of consumer goods, including both food and non-food items; Supermarket: Smaller self-service retail markets compared to hypermarkets but larger than traditional grocery stores, primarily selling food and household goods; Premium Supermarket: Upscale versions of typical

supermarkets, providing high-quality and specialty products such as organic, gourmet, and imported foods.

A total of 2736 PBMA products were surveyed. We determined both the total number and variability of PBMAs available in each store category. The countries of origin of the PBMA product categories were analysed. The average number of PBMAs per store category was calculated. Differences in PBMA numbers across store categories were evaluated with Analysis of Variance (ANOVA) and Tukey's Honestly Significant Difference (HSD) tests.

**Price analysis.** Prices per 100g of packing size were calculated to standardize product prices. Six items lacking price or packaging size data were excluded, resulting in a final sample size of $n = 2730$. Price data were reported in MYR by product category, and price variations across store types were analysed.

**Energy and nutrient content comparison.** Due to the unavailability of saturated fat information in the Malaysian Food Composition Database (MyFCD) required for nutrient profiling, data for meat equivalents were sourced from Singapore's Energy & Nutrient Composition Database (https://focos.hpb.gov.sg/eservices/ENCF/), due to its geographical proximity, cultural similarity, and the fact that it imports most of its food products from Malaysia. The meat equivalent products ($n = 109$) were selected to correspond with the PBMA product categories. Each variable (energy, protein, total fat, saturated fat, carbohydrates, and sodium) was analysed across all product categories. To evaluate the differences in the energy and nutrient contents between PBMAs and their meat equivalents, two-sample $t$ tests were conducted.

**Nutrient profiling.** PBMAs were assessed using the Nutri-Score Nutritional Rating System, developed by the French Public Health Agency (Santé Publique France) [26]. This scoring system was selected as it is a widely used front-of-pack labelling system that provides user-friendly information on the nutritional quality of food and beverages. This food-labelling system has been applied voluntarily in food products sold in European Union countries such as Sweden, Italy, and Germany; hence, nutrient profiling of PBMAs in these countries also utilised Nutri-Score [21–23]. The Nutri-Score grading and colour scheme for solid foods are as follows: A (Most Healthy; Dark Green): -15 to -1 points; B (Light Green): 0 to 2 points; C (Yellow): 3 to 10 points; D (Orange): 11 to 18 points; E (Least Healthy; Red): 19 to 40 points [27]. The Nutri-Score calculation is based on an algorithm considering positive nutritional factors (fruits, vegetables, legumes, fibre, protein) and negative factors (energy, sugars, saturates, salt). Based on the definition of PBMAs as stated in section 2.1, the fruits, vegetables, and legumes content for all PBMAs were set at 100%. The Nutri-Score was reported as mean ± standard deviation (SD) by product category, and the frequencies of each Nutri-Score colour were reported for each product category (S4 Table).

## Statistical analysis

Data extracted from the images were compiled into a Google Sheet, including product name, brand, price (MYR), packing size (g), product origin, energy (kcal/100g), and nutritional content per 100g (protein, total fat, saturated fat, cholesterol, carbohydrates, total sugars, dietary fibre, and sodium). All statistical tests were conducted in the R statistical environment, v4.0.3. Normality was assessed using the Shapiro-Wilk test. All hypothesis testing was two-sided, and a $p$-value $< 0.05$ was considered statistically significant.

## Results

### Availability of PBMAs

Across 81 surveyed supermarkets and hypermarkets in Petaling District, 34 were hypermarkets, 10 supermarkets, and 37 premium supermarkets. A total of 2736 PBMA product items

were available from these 81 surveyed locations, with 251 being unique products manufactured by 42 brands. The most prevalent PBMA product category was Coated Meats, followed by Pieces/Chunks/Fillets/Strips, and Pastries (Fig 1A). Luncheon Meat, Minced Meat, and Sausages were the least available PBMA product categories (Fig 1A). Premium supermarkets offered the highest variety of PBMAs across all product categories, with the mean of 41.2 products per store (Fig 1B). This mean was significantly higher than hypermarkets (Mean: 25.4; $p < 0.001$), but not supermarkets (Mean: 34.6; $p = 0.47$). Of the identified PBMAs, 73.71% were locally produced, while 26.29% were imported from 8 countries: Thailand, Hong Kong, Singapore, Australia, the United Kingdom, Korea, China, and the United States (Fig 1C). Malaysia dominated the origins of PBMAs across all categories, especially in Pieces/Chunks/Fillets/Strips. Most product categories feature PBMA items originating from 2 or more additional countries besides Malaysia, except for Luncheon Meat (Malaysia and Hong Kong), and Sausages (Malaysia and UK) (Fig 1C).

## Price analysis of PBMAs

Fig 2A presents the prices of PBMA product categories in terms of prices per 100g in MYR. The median prices per 100g for all categories were below MYR 7 (~USD 1.6), with Pieces/Chunks/Fillets/Strips being the most affordable category with a median price of approximately MYR 3 (~USD 0.7) per 100g. Burger/Patties and Minced Meat were the least affordable, slightly above MYR 6 (~USD 1.3) per 100g. The category with the highest price variability was Seafood Balls/Cakes/Meatballs, indicated by the widest IQR. Price comparison between store types (Fig 2B–2J) showed that supermarkets and hypermarkets generally had lower and more consistent prices, while premium supermarkets had significantly higher median prices than supermarkets and hypermarkets for Burger/Patties (MYR 7.7 or ~USD 1.8), Coated Meat (MYR 6.0 or ~USD 1.4), Minced Meat (MYR 7.5 or ~USD 1.8), Pieces/Chunks/Fillets/Strips (MYR 4.4 or ~USD 1), and Seafood Balls/Cakes/Meatballs (MYR 8.6 or ~USD 2)(Fig 2B, 2C, 2E, 2G, and 2J, respectively).

## Energy and nutrient content comparison of PBMAs

Out of 251 unique PBMAs, 35 or 14% lacked nutritional labels and were therefore excluded from the analysis, resulting in a final total of 216 PBMAs. Fig 3 and S3 Table compare the energy density and composition of key nutrients (protein, total fat, saturated fat, carbohydrates, and sodium) of PBMAs and their meat equivalents across different product categories. A total of 5 out of 9 PBMA categories had lower mean energy density than their meat equivalents, with 2 significantly lower (Pastries, Sausages; Fig 3A). On the other hand, Pieces/Chunks/Fillets/Strips had significantly higher energy density than their meat equivalents (Fig 3A). A total of 7 out of 9 PBMA categories had lower protein content than their meat counterparts, with 4 significantly lower (Luncheon Meat, Minced Meat, Pieces/Chunks/Strips, Prepacked Meals; Fig 3B). Fat content-wise, 7 out of 9 PBMA categories had lower contents, with 2 significantly lower (Pastries, Sausages; Fig 3C). A total of 6 out of 9 had lower saturated fat content compared to their meat equivalents, with 2 significantly lower (Pastries, Sausages; Fig 3D). Pieces/Chunks/Fillets/Strips had significantly higher saturated fat than its meat equivalents (Fig 3D). Notably, Coated Meat, Luncheon Meat, and Prepacked Cooked Meals had similar saturated fat levels with their meat equivalents. Conversely, 8 out of 9 PBMA categories had higher carbohydrate content, with 4 significantly higher (Burger/Patties, Pieces/Chunks/Fillets/Strips, Prepacked Meals, Seafood Balls/Cakes/Meatballs; Fig 3E). A total of 5 out of 9 had lower sodium content compared to their meat equivalents, with 3 significantly lower (Luncheon Meat, Sausages, Seafood Balls/Cakes/Meatballs; Fig 3F). In contrast, PB Pieces/Chunks/

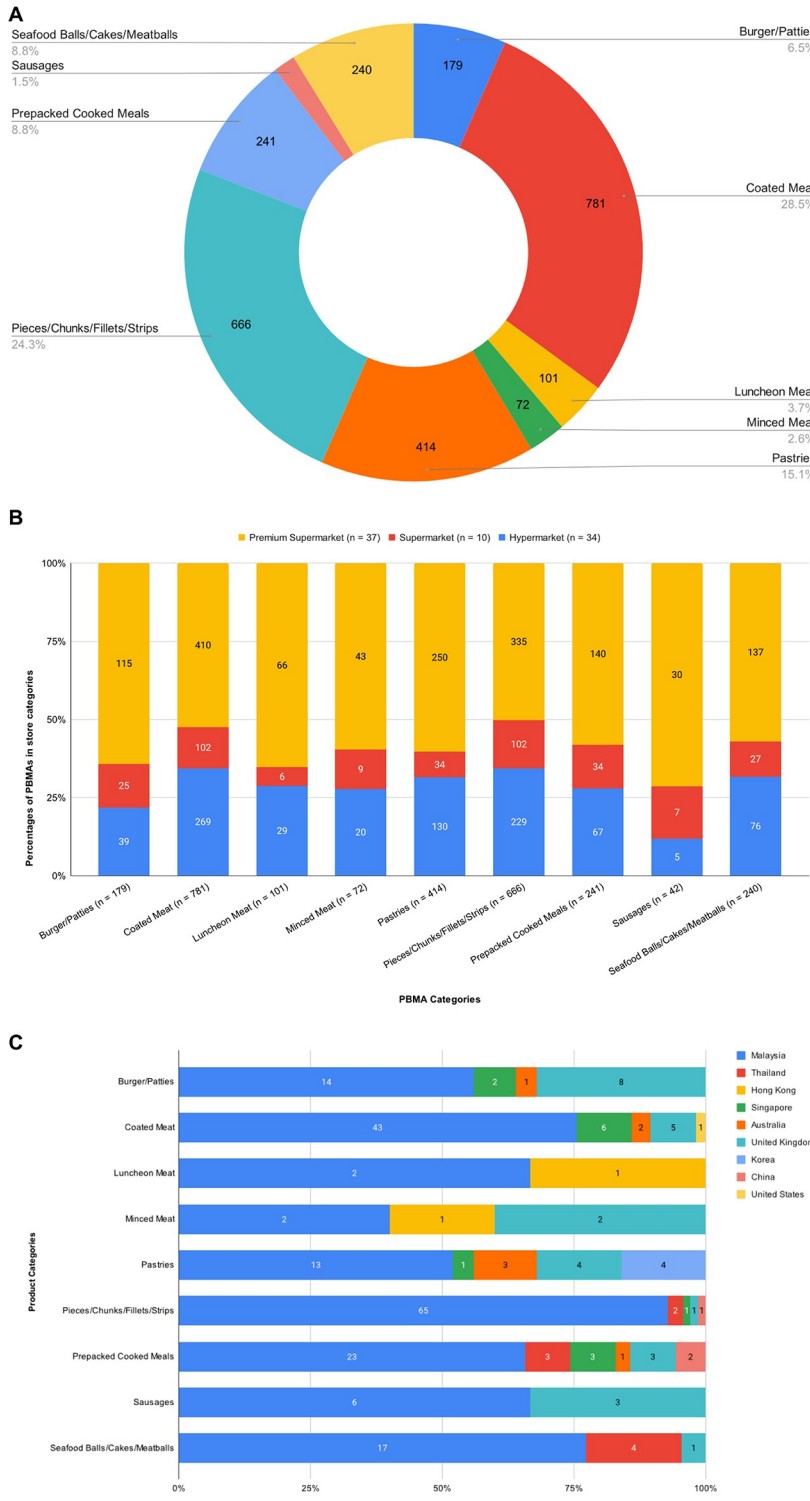

**Fig 1. Availability and origins of PBMA product categories sold in supermarkets and hypermarkets in Petaling District, Malaysia.** A. The percentages of PBMA product categories; B. The distribution of PBMA product categories across different store types: hypermarket, supermarket, and premium supermarket; C. The distribution of countries of origin for PBMA product categories.

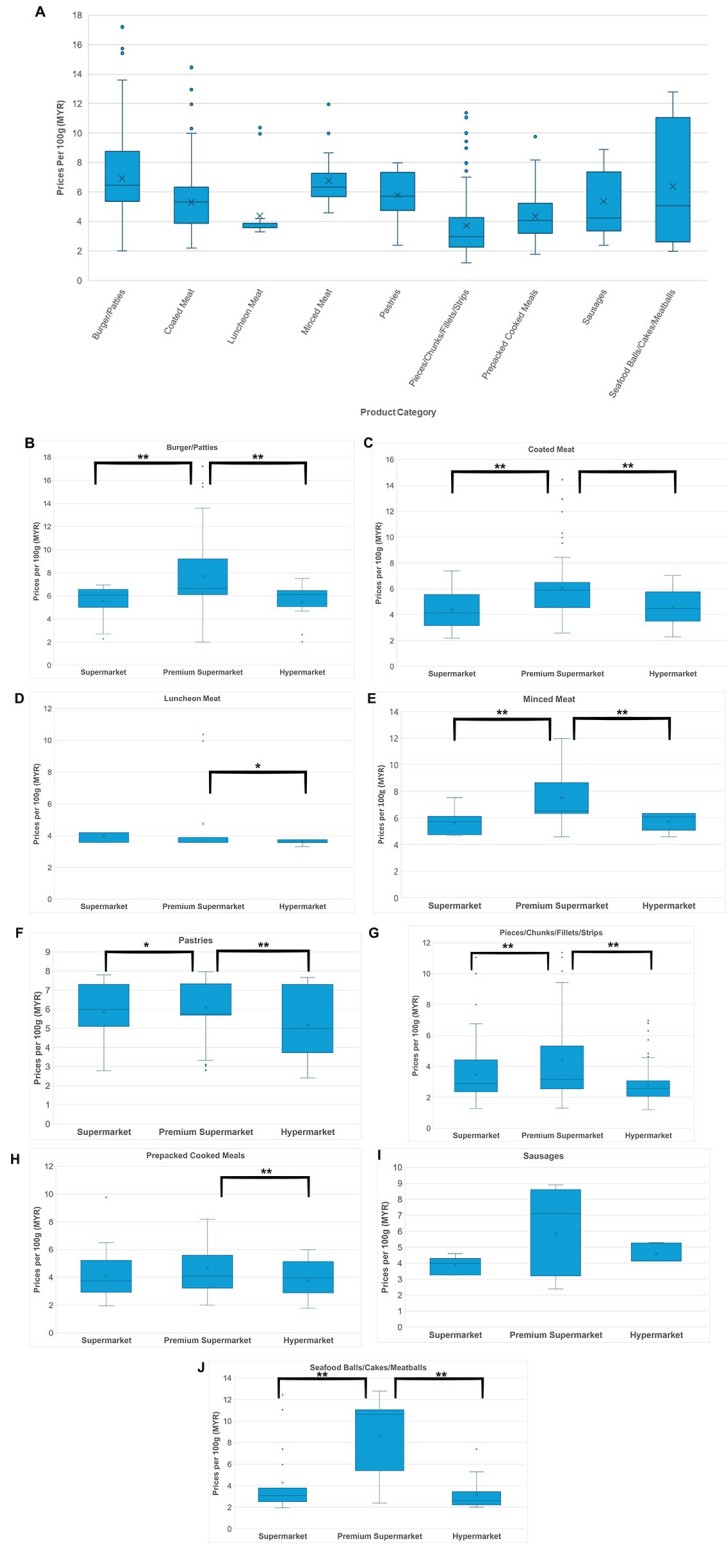

**Fig 2. Price analysis of PBMA product categories sold in supermarkets and hypermarkets in Petaling District, Malaysia.** A. Boxplot of the median prices of PBMA product categories per 100g (MYR). Boxplot comparison of the median prices of PBMA product categories per 100g (MYR) across three store types: hypermarkets, supermarkets, and premium supermarkets: B. Burger/Patties; C. Coated Meat; D. Luncheon Meat; E. Minced Meat; F. Pastries; G. Pieces/Chunks/Fillets/Strips; H. Prepacked Cooked Meals; I. Sausages; J. Seafood Balls/Cakes/Meatballs. The left and right

sides of the box are the lower and upper quartiles; the box covers the Interquartile Range (IQR); median is represented by the vertical line that split the box in two; cross represents the mean; whiskers at the bottom of box represents lower quartile, whiskers at the top of box represents upper quartile. * indicates $p < 0.05$, ** indicates $p < 0.01$; by Mann-Whitney $U$ test.

Fillets/Strips had significantly higher sodium content than their meat equivalents (Fig 3F). Overall, there was a wide variation in the energy density and nutrient content per 100g among all PBMA and meat categories.

## Nutri-score evaluation of nutritional quality of PBMAs

A total of 46 PBMAs (21.3% of 216 products having nutritional labels) that were missing all the required nutritional factors were excluded from the analysis (final $n$ = 170). The mean Nutri-Score across all PBMA categories averaged 2.96, placing them in Grade C (Yellow), indicating moderate nutritional quality. Minced Meat had the lowest mean NutriScore at -9.25, placing it in the "healthiest" category (Grade A). Sausages and Pastries fell into Grade B. The remaining categories ranged from Nutri-Scores of 3 to 10, predominantly falling under Grade C, which denotes moderate nutritional quality. Among the categories, Prepacked Cooked Meals had the highest mean Nutri-Score at 7.88, indicating the least healthy among PBMAs. Upon closer examination of individual products within each category (Fig 4 and S4 Table), Coated Meat, Pieces/Chunks/Fillets/Strips, and Prepacked Cooked Meals contained products classified under Grade E, although these represent less than 10% of the total products in each category. Most categories included products classified under Grade A, except for Luncheon Meat, where 70% of its products were Grade C and 30% Grade D.

## Discussion

Despite numerous studies assessing the nutritional profile and healthiness of PBMAs, this study is the first to offer a quantitative analysis of the availability, price, nutritional composition, and nutritional quality of PBMAs in Malaysian supermarkets and hypermarkets. Our survey indicates a significant presence of PBMAs in Petaling District, with over 95% of visited supermarkets carrying these products. This aligns with global trends of increasing PBMA consumption, reflecting their acceptance and integration into both vegetarian/vegan and omnivorous diets [13]. Notably, premium supermarkets had greater PBMA variety compared to hypermarkets and regular supermarkets, primarily because they cater to a more affluent and health-conscious customer base, driving demand for a broader selection of PBMAs. However, all PBMA categories were still available in hypermarkets and supermarkets, indicating decent accessibility across store types. Although PBMAs from nine countries were available, most products were produced locally in Malaysia. This is likely due to the advantages of local production, such as reduced transportation costs and import duties, leading to more competitive pricing and better alignment with local taste preferences and dietary habits [28].

The median prices per 100g for all PBMA categories were below MYR 7 (~USD 1.6). In 2023, Malaysians consumed approximately 50 kilograms of poultry meat per person [29]. Poultry, especially chicken, is the most popular meat in the country due to its affordability and versatility [30]. The ceiling price of poultry in Malaysia is MYR 1.14 (~USD 0.3) per 100g [31] and therefore, PBMAs are at least twice more expensive compared to the median price of the least costly category—Pieces/Chunks/Fillets/Strips. Categories such as Burger/Patties and Minced Meat were the most expensive, possibly due to a high percentage of imported products. As expected, premium supermarkets catering to a higher-income demographic, exhibited higher median prices and greater price variability. A study in Canada reported PBMA prices at

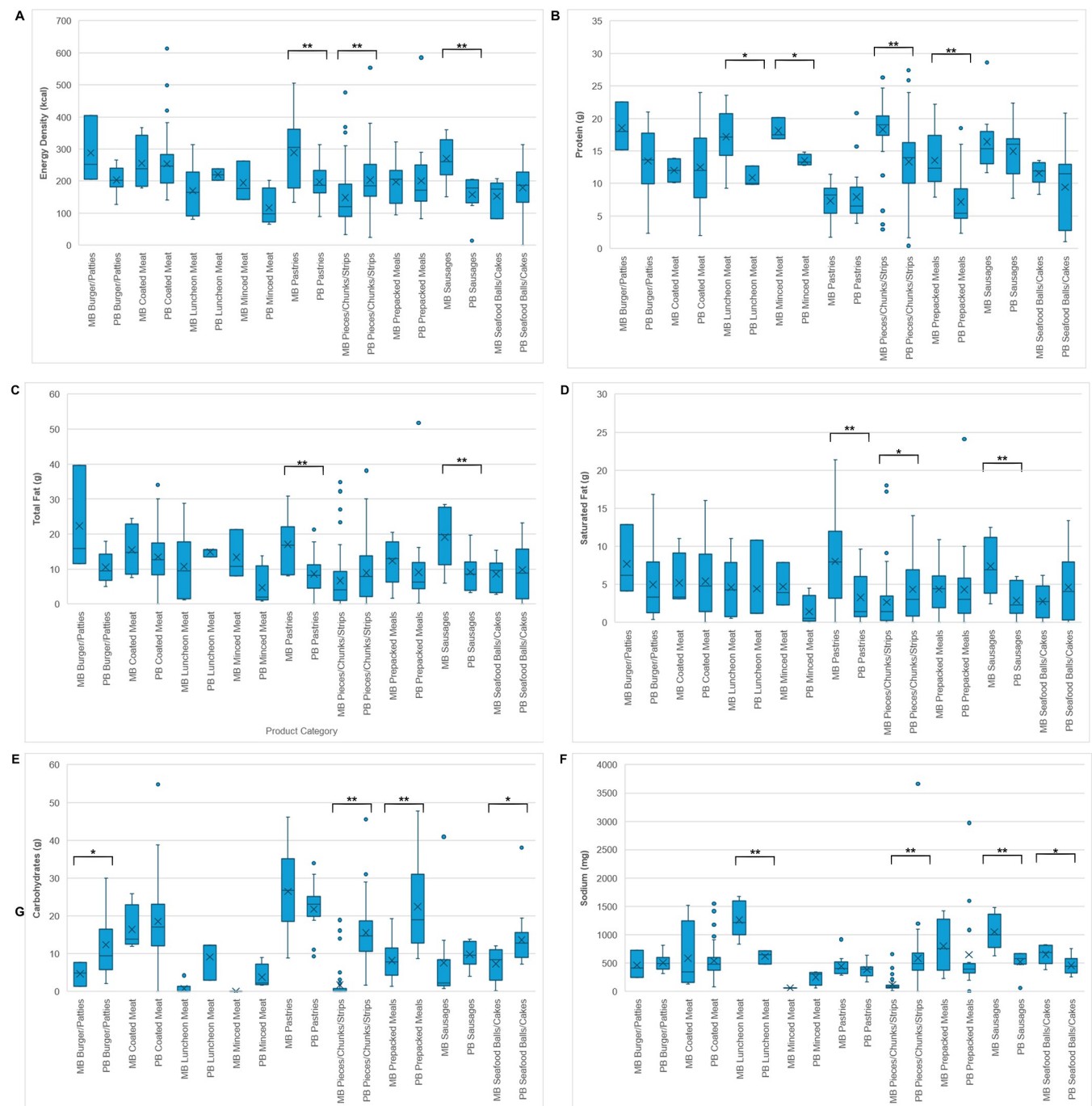

**Fig 3. Energy and nutrient content comparison of PBMA product categories sold in supermarkets and hypermarkets in Petaling District, Malaysia with meat-based equivalent product categories.** A. Energy Density (kcal); B. Protein (g); C. Total Fat (g); D. Saturated Fat (g); E. Carbohydrates (g); and F. Sodium (mg). PB: plant-based, MB: meat-based. Energy and nutrient contents were sourced from labels of PBMA products, or from the Singapore's Energy & Nutrient Composition Database for the meat-based equivalent products. * indicates $p < 0.05$, ** indicates $p < 0.01$; by $t$-test.

USD 1.95 per 100g, with no significant price difference with their meat equivalents (USD 1.81 per 100g); however, the price of plant-based cheese and yogurt analogs was higher than their animal-based counterparts [9]. In 2022, a market survey found that plant-based beef alternatives were priced higher at approximately USD 6.70/pound (or USD 1.48/100g), while beef

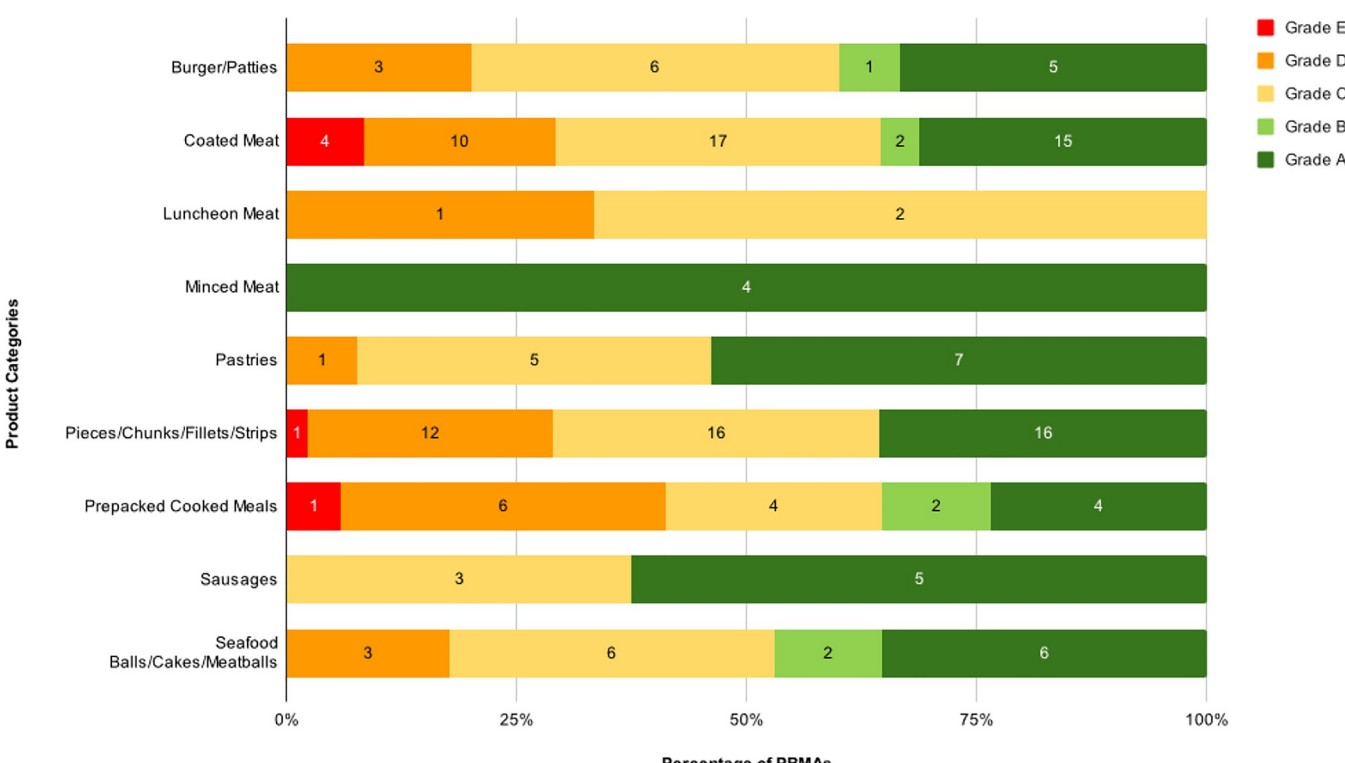

**Fig 4. Nutri-Score nutritional quality analysis of PBMA product categories sold in supermarkets and hypermarkets in Petaling District, Malaysia.** The Nutri-Score grades, colours, and points are as follows: A (most healthy; dark green): -15 to -1 points; B (light green): 0 to 2 points; C (yellow): 3 to 10 points; D (orange): 11 to 18 points; E (least healthy, red): 19 to 40 points.

costs about USD 5/pound (or USD 1.10/100g) [32]. Also, New Food Innovation reported that plant-based meat remains 67% more expensive than animal meat in the UK [33]. Compounded by the ongoing cost of living crisis and rising inflation, this price parity poses a significant barrier for consumers looking to choose more sustainable alternatives, making it even harder for people to make the switch. Indeed, an unpublished thesis reported that among 227 Malaysian respondents, 44.4% stated expensive prices as one of the reasons for not purchasing PBMAs [17]. Nevertheless, despite the higher cost of plant-based burgers, consumer surveys indicate a willingness to pay more for these items compared to other PBMA categories like sausages, chicken, and fish [34].

When comparing plant-based meat alternatives (PBMAs) in Malaysia with those in Europe (S5 Table), both have similar energy densities, meaning they offer comparable caloric content, with slight differences depending on the product category. However, Malaysian PBMAs generally have lower energy densities than European meat-based products, which could benefit consumers aiming to reduce their calorie intake. In terms of protein content, Malaysian PBMAs consistently contain less protein than both European PBMAs and meat-based products, highlighting an area for potential improvement. On the other hand, Malaysian PBMAs tend to have similar or lower total fat content compared to their European counterparts. However, they often contain higher levels of saturated fat. Compared to meat-based products, Malaysian PBMAs generally have lower total and saturated fat content, although some categories still require optimization. Malaysian PBMAs also typically contain more carbohydrates than European PBMAs and meat-based products, likely due to the use of additional fillers and binding agents, which may impact the product's nutritional balance and glycaemic load. Additionally,

Malaysian PBMAs tend to have higher sodium levels than European PBMAs, although their sodium content is generally lower than that of meat-based products, despite still being relatively high.

Over half of the PBMA categories in this study exhibited lower energy density compared to their meat equivalents, consistent with findings from other countries [5, 9, 11, 22]. This is likely due to PBMAs' lower fat and higher dietary fibre content, as they are often made from legumes and cereals [4]. Given the global obesity epidemic is linked to excessive caloric intake, PBMAs could potentially aid in weight management and reduce obesity rates [23]. Supporting this, a randomized crossover trial showed significant weight loss when participants consumed PBMAs instead of meat [35]. Additionally, consuming low energy density foods can help prevent secondary diseases such as cardiovascular diseases (CVDs) and cancer [23].

The lower total fat and saturated fat contents of PBMA categories compared to their meat equivalents is due to ingredients of the former primarily coming from vegetable oils like soybean, sunflower, and olive oils containing polyunsaturated and monounsaturated fats [5]. Although lower, excessive PBMA consumption could still lead to high fat diets, potentially affecting gut microbiota and faecal metabolic profiles [13]. In Malaysia, the average daily intake of saturated fat exceeds the recommended 10% of total daily energy intake [36]. The World Health Organization recommends reducing saturated fat intake due to its effects on low-density lipoprotein cholesterol [11]. Therefore, substituting PBMAs for meat could be a strategy to lower saturated fat intake and reduce CVD risks.

Except for Pieces/Chunks/Fillets/Strips, the Malaysian PBMAs surveyed in this study had lower sodium content than their meat equivalents, contradicting with previous studies in other countries which found higher sodium content in the former [5, 11, 23, 37]. Following the UK's traffic light labelling scheme for sodium content [38], many PBMA products analysed here would fall under the medium sodium category, with levels typically ranging from 120 mg to 600 mg per 100g. While some product categories, like Luncheon Meat and Prepacked Cooked Meals may exceed this range, the majority exhibit moderate sodium content. Despite variations, it is essential to note that processed foods, both plant-based and meat-based, often contain elevated sodium levels to enhance taste and preserve shelf life [5]. Given the World Health Organization's recommendation of limiting sodium intake to 2300 mg per day, a single 100g serving of certain PBMA categories could contribute up to 22% of this recommended daily limit. As the popularity of PBMAs grows, there is a need for manufacturers to prioritize sodium reduction initiatives and refine product formulations, potentially targeting levels that qualify for the low sodium category (<120 mg per 100g) to align with health guidelines and consumer expectations.

While PBMAs were generally favourable in terms of energy, total fat, saturated fat, and sodium contents, more than half of the categories had lower protein levels compared to their meat equivalents. Meat is a primary source of protein, comprising nearly 22% of its composition, whereas plant-based products often have lower protein levels and may lack certain essential amino acids [5, 23]. This poses a challenge for those on meatless diets, highlighting the need for health professionals to address protein intake concerns and develop public health policies accordingly. Nonetheless, plant protein intake has been associated with a reduced risk of type 2 diabetes, cardiovascular diseases, and overall mortality [23].

All PBMA categories, except Pastries, had higher carbohydrate content than their meat equivalents. This aligns with other studies reporting higher carbohydrates in PBMAs [5, 22–24, 37]. However, the higher carbohydrate content may not negatively impact diet quality. A study on a plant-based diet rich in carbohydrates from whole grains and legumes and low in fat found it effective for weight loss and improving quality of life [39]. Therefore, the carbohydrate content in PBMAs, derived from legumes and cereals, may not be excessive in a balanced diet.

Malaysian PBMAs may be suited for individuals, especially those with high cholesterol and hypertension, who seek to lower their fat and sodium intake. These products typically have lower levels of total fat, saturated fat, and sodium compared to meat-based alternatives, which can aid in managing cholesterol and blood pressure, thereby supporting cardiovascular health. Research indicates that vegetarian diets are linked to a lower risk of all-cause mortality, and specifically reduce the risk of cardiovascular and coronary artery diseases [40]. However, Malaysian PBMAs may not be ideal for those following high-protein, low-carbohydrate diets such as the Atkins or Keto diets [41]. Due to their lower protein content and higher carbohydrate levels, these products may not align with the needs of those on low-carbohydrate diets, which are often used therapeutically for conditions like severe obesity and type 2 diabetes, with proven benefits for weight loss and metabolic control in the short to medium term [41]. To significantly increase the protein content of Malaysian PBMAs, food manufacturers should diversify and intensify protein-rich ingredients like soy protein isolate, pea protein, lentils, chickpeas, beans, textured vegetable protein, wheat gluten, hemp seeds, pumpkin seeds, almonds, and mycoprotein [42] into their formulation.

The Nutri-Score was used to assess the healthiness of PBMA categories, similar to other studies [18, 21–23]. The Nutri-Score, a common front-of-package label, has sparked debate over its effectiveness in evaluating nutritional quality [43]. The Malaysian PBMAs surveyed in this study had Nutri-Score ranging from Grade A to Grade C, with Prepacked Cooked Meals having the highest mean, likely due to their higher energy density and sodium content from added oils and sauces. The Nutri-Score, while useful, does not account for the degree of food processing. Highly processed foods and junk foods are typically calorie-dense but nutrient-poor. They are engineered for flavour, often using refined sugars, high-fructose corn syrup, unhealthy trans fats, and a host of preservatives [44]. The long-term consumption of these foods can lead to significant health problems, including obesity, CVD, diabetes, cancer, and poor mental health [44]. In contrast, semi-processed foods and whole grains—such as whole-wheat bread, brown rice, quinoa, oats, and minimally processed vegetables [45]—are rich in essential nutrients like fibre, vitamins, and minerals. These foods are digested more slowly, resulting in a steadier supply of energy and numerous health benefits like weight management, heart health, improved digestive health, lower risk of chronic diseases, and better mental health [46, 47]. Therefore, Nutri-Score should be supplemented with labels indicating processing levels, such as the NOVA classification. A recent survey in Spain found that 37–41% of PBMAs studied were classified as ultra-processed or NOVA Group 4 [18], while another survey in Portugal classified all PBMAs as ultra-processed [48]. However, critics argue that the NOVA system is simplistic and unfairly classifies PBMAs as ultra-processed, as it does not adequately evaluate the nutritional attributes of meat and dairy alternatives based on soy [49]. They argue that despite their classification as ultra-processed, PBMAs compare well with their animal-based counterparts, which are classified as unprocessed or minimally processed foods [49]. They also emphasized that it is crucial not to automatically equate "ultra-processed" with poor nutritional quality, as the quality of food is determined by its final composition rather than just the level or complexity of processing [49]. Regardless of the classification of processing of PBMAs, we opine that it is important to have a balanced diet by complementing PBMAs with whole grains and unprocessed/semi-processed vegetables and fruits for even better health benefits.

Nevertheless, this study had some limitations and uncertainties. First, the nutritional data of meat equivalents sourced from Singapore's Energy & Nutrient Composition Database was mostly based raw or uncooked data, whereas PBMAs surveyed in this study were mainly processed or cooked. This difference in preparation methods may have influenced the comparative nutritional analysis. A direct comparison could be performed in the future by surveying

equivalent PBMA product items that are meat-based. Secondly, the Nutri-Score calculation assumed a uniform 100% content of fruits, vegetables, legumes, nuts, and oils across all PBMAs, potentially leading to an overestimation of their nutritional quality. The study did not include an evaluation of micronutrient content. Micronutrients, including vitamins and minerals such as zinc and iron, are crucial for overall health, and omitting their assessment limits a comprehensive understanding of the nutritional profile of PBMA products. Lastly, the cross-sectional nature of this study makes us unable to evaluate the availability and pricing of PBMAs over time, especially as consumer demand increases. Hence, follow-up studies–like 6 months or 1 year later—are needed.

## Conclusions

In summary, our results demonstrated that PBMAs were widely available across most supermarkets and hypermarkets in the Petaling District, though the variety and number of products vary by store. The price analysis indicated that median prices for PBMA categories were generally below MYR 7 (~USD 1.6) per 100g, with Burger/Patties and Minced Meat identified as relatively less affordable options. Nutritionally, PBMAs showed lower levels of energy, total fat, saturated fat, sodium, and protein compared to their meat equivalents, while exhibiting higher carbohydrate content. The Nutri-Scores assigned to these categories ranged from -15 to 10 (Grade A to C), indicating generally moderate nutritional profiles without falling into the "less healthy" orange or red zones. Considering these findings, PBMAs present themselves as a viable choice for consumers in Malaysia who are concerned about healthier and more sustainable food options, in terms of availability, price, and nutritional composition (albeit with the caveat of lower protein and higher carbohydrate). However, further research should include direct comparisons with local meat equivalents, encompassing both processed and unprocessed varieties, to better understand the complete nutritional benefits and limitations of PBMAs. Additionally, a thorough assessment of micronutrient contents, especially zinc and iron, are essential to fully assess the health implications of these products, ensuring that consumers are well-informed about their dietary choices. To improve awareness of the nutritional quality of PBMAs in Malaysia, a voluntary front-of-pack labelling system should be encouraged for all PBMA products.

## Supporting information

**S1 Table. PBMA product categories and their descriptions.**
(DOCX)

**S2 Table. Distribution of PBMAs across visited supermarkets and hypermarkets in Petaling district, Malaysia.**
(DOCX)

**S3 Table. Mean energy density and nutritional contents (per 100g) ± standard deviations (Range) of plant-based vs. meat-based product categories.**
(DOCX)

**S4 Table. Nutri-score classification across PBMA product categories.**
(DOCX)

**S5 Table. Mean energy density and nutritional contents (per 100g) of Malaysian PBMAs *vs.* European PBMAs and meat equivalents.**
(DOCX)

## Author Contributions

**Conceptualization:** Yee-How Say.

**Data curation:** Katty Jia Qi Lou, Yee-How Say.

**Formal analysis:** Katty Jia Qi Lou, Nadia Nantheni Rajaram, Yee-How Say.

**Investigation:** Katty Jia Qi Lou, Yee-How Say.

**Methodology:** Yee-How Say.

**Project administration:** Yee-How Say.

**Supervision:** Yee-How Say.

**Validation:** Katty Jia Qi Lou, Nadia Nantheni Rajaram, Yee-How Say.

**Visualization:** Katty Jia Qi Lou, Yee-How Say.

**Writing – original draft:** Katty Jia Qi Lou, Nadia Nantheni Rajaram, Yee-How Say.

**Writing – review & editing:** Nadia Nantheni Rajaram, Yee-How Say.

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
