## [Decision Letter · Decision Letter 0]

4 Oct 2024

PONE-D-24-34533Availability, Price and Nutritional Assessment of Plant-Based Meat Alternatives in Hypermarkets and Supermarkets in Petaling, the Most Populated District in MalaysiaPLOS ONE

Dear Dr. Say,

Thank you for submitting your manuscript to PLOS ONE. After careful consideration, we feel that it has merit but does not fully meet PLOS ONE’s publication criteria as it currently stands. Therefore, we invite you to submit a revised version of the manuscript that addresses the points raised during the review process.

You can find the reviewer comments below in this email, as well as in the attachments, if available.

We look forward to receiving your revised manuscript.

Kind regards,

Karthikeyan Venkatachalam, Ph.D. in Food Science and Technology

Academic Editor

PLOS ONE

Journal Requirements:

Reviewers' comments:

Reviewer's Responses to Questions

**Comments to the Author**

1. Is the manuscript technically sound, and do the data support the conclusions?

Reviewer #1: Yes

Reviewer #2: Yes

Reviewer #3: Yes

Reviewer #4: Yes

Reviewer #5: Yes

2. Has the statistical analysis been performed appropriately and rigorously? 

Reviewer #1: N/A

Reviewer #2: Yes

Reviewer #3: Yes

Reviewer #4: Yes

Reviewer #5: N/A

3. Have the authors made all data underlying the findings in their manuscript fully available?

Reviewer #1: Yes

Reviewer #2: Yes

Reviewer #3: Yes

Reviewer #4: Yes

Reviewer #5: Yes

4. Is the manuscript presented in an intelligible fashion and written in standard English?

Reviewer #1: Yes

Reviewer #2: Yes

Reviewer #3: Yes

Reviewer #4: Yes

Reviewer #5: No

5. Review Comments to the Author

Reviewer #1: The findings of this study contribute significantly to the fields of nutrition, food science, and public health, particularly in the context of Malaysia. The research highlights the availability and nutritional profiles of PBMAs, which can inform consumers, health professionals, and policymakers about healthier dietary options. Furthermore, the study's insights into pricing can aid in understanding market dynamics and consumer behavior regarding plant-based products.

1. The study mentions that PBMAs generally have lower protein content compared to meat equivalents. What specific strategies do you suggest for improving the protein content of PBMAs in the Malaysian market?

2. Given the findings on the nutritional quality of PBMAs, how do you plan to disseminate this information to consumers to help them make informed dietary choices?

3. Have you considered conducting follow-up studies to assess changes in the availability and pricing of PBMAs over time, especially as consumer demand increases?

4. You noted the need for further research on micronutrient profiles. What specific micronutrients do you believe are most critical to assess in future studies, and why?

5. You mentioned limitations regarding the use of the Singapore database for meat equivalents. How might this impact the validity of your comparisons, and what steps could be taken in future research to address this issue?

6. The discussion touches on the health implications of ultra-processed foods. How do you propose to evaluate the degree of processing in PBMAs in future studies, and what impact do you think this has on consumer health?

7. Considering Malaysia's diverse culinary landscape, how do you think cultural preferences might influence the acceptance and consumption of PBMAs?

Overall, the manuscript presents valuable research that is timely and relevant to current dietary trends in Malaysia. Addressing the questions posed could enhance the manuscript's depth and provide further insights into the implications of the findings. I recommend minor revisions to improve clarity and conciseness before publication.

Reviewer #2: I suggest authors carefully revise the comments given at each point in the manuscript and recommend for publication in this journal. I have included my little comments along with separate files in the attachment.

Reviewer #3: First, congratulations on the comprehensive research regarding an emerging theme. Here are my minor suggestions and doubts to help the authors improve the clarity and quality of the article:

Suggestions:

Enhance the description of the rationale behind the Nutri-Score in the methodological section for better clarity.

Doubts:

Out of the total of 2,736 PBMAs, why were only 251 subject to the Energy and Nutrient Content Comparison and Nutri-Score Evaluation of Nutritional Quality analysis? Specifying the inclusion and exclusion criteria could strengthen the credibility of the analysis.

Reviewer #4: Thank you for the opportunity to review Manuscript Number PONE-D-24-34533 for the Journal of PLOS ONE. This scientific research study titled "Availability, Price, and Nutritional Assessment of Plant-Based Meat Alternatives in Hypermarkets and Supermarkets in Petaling, the Most Populated District in Malaysia" presents intriguing findings and will significantly enhance the understanding of plant-based food markets in the field of science.

Please, see attached (word, pdf) files

Reviewer #5: I have completed the review of the manuscript entitled "Availability, Price and Nutritional Assessment of Plant-Based Meat Alternatives in Hypermarkets and Supermarkets in Petaling, the Most Populated District in Malaysia”. There are several areas where the manuscript can be strengthened for clarity and scientific rigor.

Comments to the Author

1. Please define Abbreviation in first appearance. The abbr. PBMA must be defined in both the abstract and at the beginning of the introduction.

2. The article does not mention with a compelling argument for why the quantitative analysis of PBMAs in Malaysia is necessary. The narrative of rising demand in Malaysia does not convincingly signify the demand just by correlating it with broad global market trends. This leads to the broader description of problem; and strengthening this aspect will better justify the study objectives.

3. a. The study excluded small and medium-sized grocery stores, potentially missing a segment of the market that may offer different PBMA varieties or pricing structures. How will the authors address that?

b. How would you justify that selection criteria of supermarkets and hypermarkets via Google searches and on-site visits did not introduce any biases in the selection of supermarkets and hypermarkets.

c. Please correct "Supermarkets and Hypermarkets in Subang" to reflect multiple location name in order to avoid confusion that there was only one google search for Subang.

4. In the methodology describing nutritional profiling, please explain Nutri-Score system. While the color scheme and scores are mentioned, please specify a detailed explanation for -15 (healthiest) to +40 (least healthy), similar to the caption of figure 4, for clarity and context.

5. Missing “Full stop” in line 204 and line 228.

6. Line 280-292, supporting evidence for price comparisons between PBMA provides contradictory findings. Even though the PBMA prices in Canada are similar to those in Malaysia, the cost of meat in Canada is not significantly different than Canadian PBMA. This undermines the argument that PBMA are expensive. It is recommended to include additional references to further support the price comparison.

Furthermore, please discuss how the price variations across regions or product categories affects consumer behavior, if possible.

7. The PBMAs generally have lower energy densities due to factors like lower fat, higher water, fiber content, and hydrocolloid binders (replacement of fat that yields more energy) compared to meat products. However, the comparison between Malaysian PBMAs and European meat products (lines 296) is ambiguously presented. The phrase “Malaysian PBMAs generally have lower energy densities than European meat-based products” suggests that European PBMAs may have higher energy density, which could be misleading. ……In fact, rewriting this whole paragraph would do the justice to the manuscript (Rewrite line 294-308).

6. PLOS authors have the option to publish the peer review history of their article (what does this mean?). If published, this will include your full peer review and any attached files.

Reviewer #1: No

Reviewer #2: No

Reviewer #3: **Yes: **Celson Júnio do Nascimento Costa

Reviewer #4: **Yes: **Woroud Alsanei

Reviewer #5: **Yes: **Prashant Dahal

---

## [Author Response · Author response to Decision Letter 0]

7 Oct 2024

The Response to Reviewers are appended in the file "PLOSONE PBMA Response to Reviewers".

---

## [Decision Letter · Decision Letter 1]

19 Oct 2024

PONE-D-24-34533R1Availability, Price and Nutritional Assessment of Plant-Based Meat Alternatives in Hypermarkets and Supermarkets in Petaling, the Most Populated District in MalaysiaPLOS ONE

Dear Dr. Say,

Thank you for submitting your manuscript to PLOS ONE. After careful consideration, we feel that it has merit but does not fully meet PLOS ONE’s publication criteria as it currently stands. Therefore, we invite you to submit a revised version of the manuscript that addresses the points raised during the review process.

The reviewers appreciated your efforts in revising the original version of the manuscript, and several have recommended accepting your work. However, there are still a few minor comments from Reviewer 2. I kindly request that you address these comments before I finalize the decision on your manuscript.

We look forward to receiving your revised manuscript.

Kind regards,

Karthikeyan Venkatachalam, Ph.D. in Food Science and Technology

Academic Editor

PLOS ONE

Journal Requirements:

Reviewers' comments:

Reviewer's Responses to Questions

**Comments to the Author**

1. If the authors have adequately addressed your comments raised in a previous round of review and you feel that this manuscript is now acceptable for publication, you may indicate that here to bypass the “Comments to the Author” section, enter your conflict of interest statement in the “Confidential to Editor” section, and submit your "Accept" recommendation.

Reviewer #2: (No Response)

Reviewer #5: All comments have been addressed

2. Is the manuscript technically sound, and do the data support the conclusions?

Reviewer #2: Yes

Reviewer #5: Yes

3. Has the statistical analysis been performed appropriately and rigorously? 

Reviewer #2: Yes

Reviewer #5: Yes

4. Have the authors made all data underlying the findings in their manuscript fully available?

Reviewer #2: (No Response)

Reviewer #5: Yes

5. Is the manuscript presented in an intelligible fashion and written in standard English?

Reviewer #2: (No Response)

Reviewer #5: Yes

6. Review Comments to the Author

Reviewer #2: In the discussion section:

1. I suggest that the paper will be more insightful if you can highlight the narrative regarding the effect of processed or junk foods compared to semi-processed or whole-grain foods on health with supporting from existing literature.

2. Again, the results presentation in the discussion section seems to be a duplication; rather, stick to the possible reasons and consequences, and comparison of the finding with previous works is important.

Reviewer #5: The authors have properly amended the manuscript by considering the reviewers' comments. The revised manuscript is now acceptable.

7. PLOS authors have the option to publish the peer review history of their article (what does this mean?). If published, this will include your full peer review and any attached files.

Reviewer #2: No

Reviewer #5: **Yes: **Prashant Dahal

---

## [Author Response · Author response to Decision Letter 1]

20 Oct 2024

1. I suggest that the paper will be more insightful if you can highlight the narrative regarding the effect of processed or junk foods compared to semi-processed or whole-grain foods on health with supporting from existing literature.

The following narrative has been added to discussion:

• Highly processed foods and junk foods are typically calorie-dense but nutrient-poor. They are engineered for flavour, often using refined sugars, high-fructose corn syrup, unhealthy trans fats, and a host of preservatives [44]. The long-term consumption of these foods can lead to significant health problems, including obesity, CVD, diabetes, cancer, and poor mental health [44]. In contrast, semi-processed foods and whole grains - such as whole-wheat bread, brown rice, quinoa, oats, and minimally processed vegetables [45] - are rich in essential nutrients like fiber, vitamins, and minerals. These foods are digested more slowly, resulting in a steadier supply of energy and numerous health benefits like weight management, heart health, improved digestive health, lower risk of chronic diseases, and better mental health [46,47].

• Regardless of the classification of processing of PBMAs, we opine that it is important to have a balanced diet by complementing PBMAs with whole grains and unprocessed/semi-processed vegetables and fruits for even better health benefits. 

2. Again, the results presentation in the discussion section seems to be a duplication; rather, stick to the possible reasons and consequences, and comparison of the finding with previous works is important.

We have already avoided the duplication of results in the discussion section in our first revision. Nevertheless, we have further removed the repetition of results, as follows:

• “Compared to the median price per 100g of the least expensive PBMA category (Pieces/Chunks/Fillets/Strips at MYR 2.98 or ~USD 0.7), PBMAs are at least twice as expensive as chicken, which has a ceiling price of MYR 1.14 (~USD 0.3) per 100g [31]. Categories such as Burger/Patties and Minced Meat were the most expensive, possibly due to a high percentage of imported products” changed to: “The ceiling price of poultry in Malaysia is MYR 1.14 (~USD 0.3) per 100g [31] and therefore, PBMAs are at least twice more expensive compared to the median price of the least costly category - Pieces/Chunks/Fillets/Strips. Categories such as Burger/Patties and Minced Meat were the most expensive, possibly due to a high percentage of imported products.”

• “Our data also showed that most PBMA categories had lower total fat and saturated fat content compared to their meat equivalents. The fats in PBMAs primarily come from vegetable oils, such as soybean, sunflower, and olive oils, which contain polyunsaturated and monounsaturated fats, unlike the saturated fats found in greater amounts in meat [5]” rewritten to “The lower total fat and saturated fat contents of PBMA categories compared to their meat equivalents is due to ingredients of the former primarily coming from vegetable oils like soybean, sunflower, and olive oils containing polyunsaturated and monounsaturated fats [5].”

• “Most PBMA categories in this study had lower sodium content than their meat equivalents, except for Pieces/Chunks/Fillets/Strips, which had significantly higher sodium levels. This contradicts other studies that consistently found higher sodium content in PBMAs” rewritten to “Except for Pieces/Chunks/Fillets/Strips, the Malaysian PBMAs surveyed in this study had lower sodium content than their meat equivalents, contradicting with previous studies in other countries which found higher sodium content in the former”. 

• “Our results showed PBMA categories ranged from Grade A to Grade C, with more than half falling under Grade C and Nutri-Scores ranging from 3 to 10. Prepacked Cooked Meals had the highest mean Nutri-Score of 7.88, likely due to their higher energy density and sodium content from added oils and sauces: rewritten to “The Malaysian PBMAs surveyed in this study had Nutri-Score ranging from Grade A to Grade C, with Prepacked Cooked Meals having the highest mean, likely due to their higher energy density and sodium content from added oils and sauces.”

---

## [Editor Report · Decision Letter 2]

24 Oct 2024

Availability, Price and Nutritional Assessment of Plant-Based Meat Alternatives in Hypermarkets and Supermarkets in Petaling, the Most Populated District in Malaysia

PONE-D-24-34533R2

Dear Dr. Say,

We’re pleased to inform you that your manuscript has been judged scientifically suitable for publication and will be formally accepted for publication once it meets all outstanding technical requirements.

Kind regards,

Karthikeyan Venkatachalam, Ph.D.

Academic Editor

PLOS ONE
---

## [Editor Report · Acceptance letter]

14 Nov 2024

PONE-D-24-34533R2 

PLOS ONE

Dear Dr. Say, 

I'm pleased to inform you that your manuscript has been deemed suitable for publication in PLOS ONE. Congratulations! Your manuscript is now being handed over to our production team.

Kind regards, 

on behalf of

Dr. Karthikeyan Venkatachalam 

Academic Editor

PLOS ONE